# Effect of Recycled Plastic Granules as a Partial Substitute for Natural Resource Sand on the Durability of SCC

**Bala Rama Krishna Chunchu** [1] and **Jagadeesh Putta** [2,*]

1    Research scholar, Department of Structural and Geo-technical engineering, School of Civil Engineering, Vellore Institute of Technology, Vellore-632014, India
2    Professor, School of Civil Engineering, Vellore Institute of Technology, Vellore-632014, India
*    Correspondence: p.jagadeesh@vit.ac.in; Tel.: +91-944-471-2064

**Abstract:** This investigation is focused on durability studies of binary blended self-compacting concrete (SCC) with the replacement effect of electronic plastic waste, namely high-impact polystyrene (HIPS) granules as partial sand. In the current investigation, for all the SCC mixes, cement is replaced with pozzolanic material fly ash in the binder content of 497 kg/m$^3$ and an adopted water-to-binder ratio of 0.36. Durability properties such as porosity, water absorption, and sorptivity are assessed for the curing periods of 28 and 90 days on SCC specimens produced with HIPS (0%–40% replacement by volume of sand). Both surface and internal water absorption rates were found to be minimal for SCC with HIPS. Replacement of HIPS up to 30% in SCC exhibited improved trends for all tests results. Reported durability parameter values were within permissible limits and revealed the excellent performance of HIPS in SCC. The optimum durability values can be attributed to the dense microstructure of SCC obtained with the combined effect of HIPS and fly ash. The continuous gradation of aggregates in the matrix reduced porosity due to the spherical shape of HIPS; additionally, the hydrophobicity of HIPS inhibits moisture migration in SCC. The additional benefits of fly ash, such as pozzolanic action and the filler effect at the interfacial transition zone (ITZ) are also major contributions to the long-term performance of durability. Electronic plastic waste replacement for fine aggregates in concrete compensates for the disposal problem and conserves natural sand.

**Keywords:** durability; high impact polystyrene; porosity; self-compacting concrete; sorptivity; water absorption

---

## 1. Introduction

The structure's serviceability depends on the durability that maintains the engineering properties of concrete, resisting severe environmental action. Durability depends mostly on the liquid's penetration capacity, which disturbs the chemical stability in the microstructure of concrete [1]. Thus, formed porosity is the measure of water saturation in the microstructure. The inter-connected pore structure allows a passage for fluid to be involved in the hydration process [2]. Permeability depends on the pore structure, micro-cracks in the paste and at the interfacial transition zone (ITZ) [3]. However, curing conditions are the main factors for pore structure. Temperature and duration of curing are the main factors for pore structure. The initial curing period plays an important role, in durability performance mainly, in the presence of mineral admixtures like fly ash for obtaining effective pozzolanic action in the concrete [4]. Pozzolanic materials can improve the microstructure in addition to plastic waste in concrete [5]. Admixtures reduce the air content by affecting the hydrophobicity of plastic aggregates in concrete [6]. However, high volume plastic content in concrete creates a high amount of porosity [7].

It has been found that the negligible water absorption experienced up to 10% replacement of plastic aggregates compared to the reference concrete [8]. The hygroscopic properties of self-compacting concrete (SCC) have been characterized by a decrease of water vapor adsorption capacity with the increasing percentage of plastic aggregates [9]. Increase in the amount of coarse plastic aggregate results in an exponential increment of water absorption in concrete [10]. Sorptivity is an easy method to determine the surface absorption rate. Moisture migration into pores through capillary suction can be evaluated. Total volume of capillary and gel pores can be estimated by weighing the water ingress into specimens from the soaking period [10–14]. In general, sorptivity and water absorption results follow a similar trend, though there is no such kind of exact relation [10]. Therefore, it has been found in studies that plastic waste in SCC can be replaced for fine aggregates for satisfying durable performance, though strength reduces [12–16]. However, flow-ability and slump retention capacity have performed better in SCC at a lower plastic replacement for sand [15]. Addition of admixtures such as fly ash in SCC enhance the durability due to pozzolanic activity and void filling that lead to better adhesion between cement paste and plastic aggregates [17,18]. Sorptivity coefficients, gas permeability coefficients, and chloride ion penetration of SCC with plastic waste have been shown to increase [19]. Alqahtani et al. have found that resistance to chloride ingress of concrete improves with the addition of plastic aggregates [20]. Admixtures enhance electrical resistance of SCC with plastic waste [21,22]. SCC samples with a higher percentage of polyethylene terephthalate (PET) particles retain a good load-bearing capacity against sulfuric acid attack. Hence, substituting partial sand with PET particles in SCC contributes to higher durability against sulfuric acid attack [23]. SCC with plastic aggregates has been evaluated for water absorption and sorptivity tests to study the resistance capacity of steel corrosion. Carbonation depth of concrete samples has been found to be higher for a higher amount of polyvinyl chloride (PVC) content. SCC with expanded polystyrene (EPS) has lower absorption and SCC mixtures with densities higher than 2000 kg/m$^3$ have a low corrosion risk. The air exclusion ratio has been found to be lower with a lower water/binder ratio [21]. Akçaözoǧlu et al. have reported that a drying shrinkage of 56% was increased with a 100% PET aggregate [24]. Water absorption and sorptivity values of concrete with coarse plastic aggregates are higher than conventional concrete [10,11]. Though some studies are available which are related to the durability of concrete with plastic waste, research has to be developed based on influencing factors such as the shape and size of aggregates, treated plastic aggregates, curing conditions, favorable mix compositions of concrete, and the curing period, etc., [25]. Hence, the main objective of the current investigation was to examine the durability properties of spherical shaped high-impact polystyrene (HIPS) aggregates as a partial fine aggregate in binary blended SCC.

## 2. Materials and Methods

### 2.1. Materials Used for the Development of SCC

Ordinary Portland Cement 53 Grade was used in concrete for attaining good strength. Physical and chemical properties of ordinary Portland cement are explained in Tables 1 and 2. Coarse aggregates of sizes 12 mm and 10 mm were used in the ratio of 60:40 in SCC. The fineness modulus of coarse aggregates of 10 mm and 12 mm were 5.86 and 6.98, respectively. Natural river sand with a fineness modulus of 2.26 and a maximum size of 4.75 mm was partially replaced with plastic waste HIPS. All properties of aggregates and HIPS are compared in Table 3. Chemical composition of HIPS is presented in Table 4. Among the plastic wastes, HIPS is characterized by versatility, machine-ability, and excellent dimensional stability, and is often recommended for low strength structural applications, since it is economical. In India, HIPS is used in various products such as computer housing, and Audio-visual equipment parts, etc. Aforesaid plastic waste is recycled by shredding into required sizes and is utilized in food packaging, making toys, and plastic trays, etc. Its availability is in different sizes and shapes but in the current work, to replace fine aggregates, a commercially available particle size was selected in the range of 1 mm–4 mm. Class F fly ash obtained from NTTPS Ibrahimpatnam,

Vijayawada was used and its properties are explained in Table 5. Tap water was used in concrete mix preparation. Fosroc-based viscosity modifying agent and Conplast SP430 super plasticizer (sulfonated naphthalene polymer) was used and their properties are clearly mentioned in Table 6. The percentage of dry material of super plasticizer and viscosity modifying agent used was 40%. A scanning electronic microscopy image of HIPS and Energy dispersive analysis X-ray (EDAX) of HIPS are presented in Figures 1 and 2, respectively. The surface of HIPS is smooth in texture and spherical in shape, as shown in Figure 3. Methodology followed in determining durability of SCC is described in Figure 4.

**Table 1.** Physical properties of Ordinary Portland Cement (OPC) 53 grade cement.

| Properties | Test Result | Test Method | IS 12269 (1987) Requirements |
|---|---|---|---|
| Normal consistency | 31% | IS 4031 (1988)—part 4 | - |
| Initial setting time (min) | 60 | IS 4031 (1988)—part 5 | Minimum time is 30 min |
| Final setting time (min) | 320 | IS 4031 (1988)—part 5 | Maximum time is 600 min |
| Specific gravity | 3.15 | IS 4031(1988)—part 11 | - |
| Compressive strength (MPa) | | | |
| 3 days | 30.82 MPa | | 27 MPa |
| 7 days | 49.50 MPa | IS 4031 (1988)—part 6 | 37 MPa |
| 28 days | 58.34 MPa | | 53 MPa |

**Table 2.** Chemical properties of OPC 53 grade cement.

| Particulars | Percentage of Chemical Composition (%) | IS:12269-1987Recommendations |
|---|---|---|
| Lime (CaO) | 61.85 | |
| Silica ($SiO_2$) | 20.07 | |
| Iron oxide ($Fe_2O_3$) | 4.62 | |
| Alumina ($Al_2O_3$) | 5.32 | |
| Magnesia (MgO) | 0.83 | Not more than 6.0% |
| Sulfuric anhydride ($SO_3$) | 2.50 | Maximum content is 3.0% when $C_3A$ >5.0; Maximum content is 2.5% when $C_3A$ <5.0 |
| Lime saturation factor $CaO-0.7SO_3/2.8SiO_2+1.2Al_2O_3+0.65Fe_2O_3$ | 0.91 | 0.80 to 1.02 |
| Ratio of alumina/iron oxide | 1.18 | Minimum is 0.66 |
| Chloride content | 0.0028 | Maximum content is 0.1% |

**Table 3.** Properties of coarse, fine and high-impact polystyrene aggregate

| Aggregate Property | Aggregate Type | | |
|---|---|---|---|
| | Coarse Aggregate | Fine Aggregate | HIPS Aggregate |
| Specific gravity | 2.7 | 2.6 | 1.04 |
| Water absorption (%) | 0.3% | 1% | Negligible |
| Bulk density ($kg/m^3$) | 1656 | 1609 | 650 |
| Size of aggregate (mm) | 12 mm and 10 mm in 60:40 ratio | Less than 4.75 | 1 mm–4 mm |

**Table 4.** Chemical composition of HIPS.

| Element | Weight% | Atomic% |
|---|---|---|
| C K | 82.13 | 87.10 |
| O K | 14.73 | 11.73 |
| Si K | 1.32 | 0.60 |
| Ca K | 1.82 | 0.58 |
| Total | 100.00 | |

**Table 5.** Properties of fly-ash.

| Chemical Properties | | |
| --- | --- | --- |
| **Class F Fly Ash Particulars** | **Chemical Composition (%)** | **Recommendations According to ASTM C 618** |
| Silica ($SiO_2$) | 60.5 | |
| Alumina ($Al_2O_3$) | 30.8 | |
| Iron oxide ($Fe_2O_3$) | 3.6 | $SiO_2 + Al_2O_3 + Fe_2O_3 > 70$ |
| Lime (CaO) | 1.4 | |
| Magnesia (MgO) | 0.91 | |
| Sulfuric anhydride ($SO_3$) | 0.14 | |
| $K_2O + Na_2O$ | 1.1 | Maximum of 5.0 |
| Loss on ignition | 0.29 | Maximum of 6.0 |
| Physical Properties | | |
| Specific gravity | 2.2 | |
| Fineness ($m^2$/kg) | 320 | Minimum of 225 $m^2$/kg |

**Table 6.** Properties of chemical admixtures.

| Particulars | Test Results of Super-Plasticizer FOSROC Conplast SP430 | Test Results of FOSROC Viscosity Modifying Agent |
| --- | --- | --- |
| Specific gravity | 1.20 | 1.09 |
| Color | Brown | White |
| Solid content (%) | 40 | 40 |
| Quantity (%) by cementitious weight | 0.9 | 0.2 |
| Main component | Sulfonated naphthalene polymers | Poly-carboxylate ether |

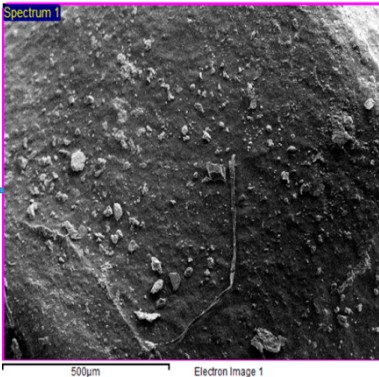

**Figure 1.** SEM image of high-impact polystyrene.

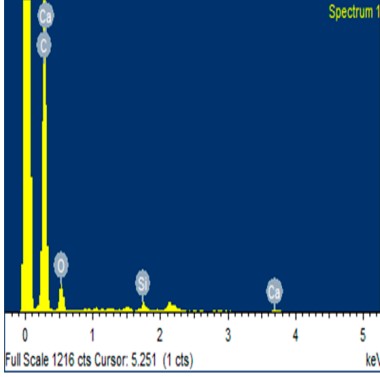

**Figure 2.** Energy-dispersive analysis X-ray (EDAX) image of HIPS.

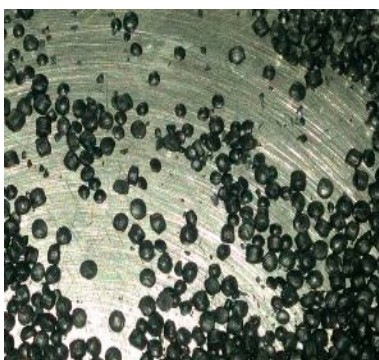

**Figure 3.** High-impact polystyrene granules.

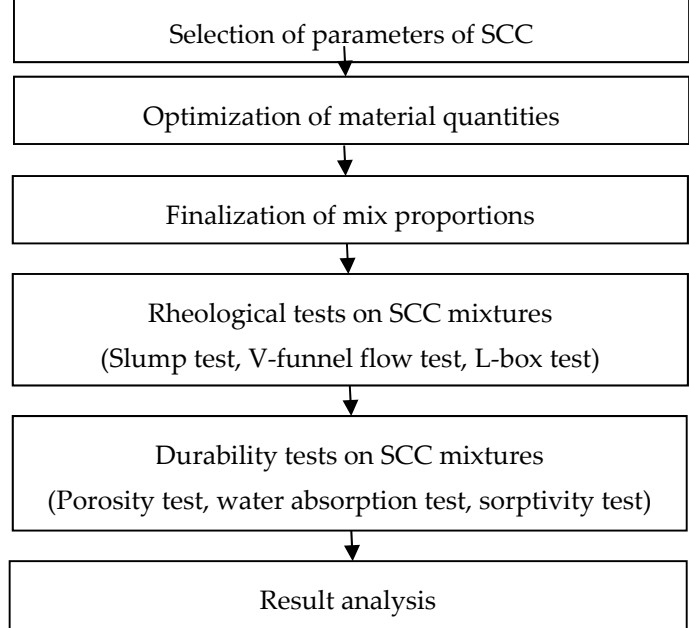

**Figure 4.** Methodology followed in determining durability of self-compacting concrete (SCC).

## 2.2. Mix Proportions

SCC was produced with binder content of 497 kg/m$^3$, replacing cement with 30% fly ash by weight and sand with HIPS (10%–40%) by volume. The fine aggregate was 54.13% (by volume). Coarse aggregates used were 12 mm of 28.08% (by weight) and 10 mm of18.72% (by weight). Water to binder ratio of 0.36 was used for all SCC mixes. SCC replaced with only 30% fly ash was used as a reference mix. SCC of M$_{30}$ grade was used to determine the durable properties. Optimized material quantities are given in Table 7. Flow-ability and mix proportions used in SCC production are shown in Table 8.

**Table 7.** Optimization of material quantities for SCC mix.

| Material Data | | | Coarse Aggregate (CA) Optimization | | | | Constituent Materials for Concrete | | | |
|---|---|---|---|---|---|---|---|---|---|---|
| Material | Specific Gravity | % Absorption | Material | | % by Weight | | Material (kg/m$^3$) | Initial | Adjusted | Per 1 m$^3$ |
| Cement | 3.15 | N/A | CA 10 mm | | 40 | | Cement | 347.90 | 347.90 | 347.90 |
| Fly ash | 2.20 | N/A | CA 12 mm | | 60 | | Fly ash | 149.10 | 149.10 | 149.10 |
| CA 12 mm | 2.70 | 0.3 | | | | | Water | 178.90 | 186.50 | 186.50 |
| CA10 mm | 2.70 | 0.3 | CA (kg/m$^3$) | | 758.44 | | Sand | 861.90 | 861.90 | 861.90 |
| Sand | 2.60 | 1.0 | % of CA | | 28.09 | | CA 12 mm | 455.00 | 455.00 | 455.00 |
| Input parameters | | Concrete mix proportions by volume (lit/m$^3$) | | Aggregate proportions | | | CA 10 mm | 303.30 | 303.30 | 303.30 |
| Dry-Rodded Unit Weight DRUW (kg/m$^3$) | 1656 | CA | 280.91 | Material | % Vol. | % Wt. | VMA (lit) | 0.99 | 0.99 | 0.99 |
| % ofCA in DRUW | 45.80 | Mortar | 719.00 | CA 12 mm | 27.50 | 28.00 | SP (lit) | 4.47 | 4.47 | 4.47 |
| % of sand | 46.10 | Sand | 331.50 | CA 10 mm | 18.30 | 18.70 | Unit wt. | 2152 | Total(kg) | 2159.60 |
| % of fly ash | 30 | Paste | 387.50 | FA | 54.10 | 53.10 | | | | |
| | | | | Total | 100 | 100 | | | | |
| Wt. water/binder | 0.36 | Total aggregates (kg/m$^3$) | | 1620.35 | | | | | | |
| Binder (kg/m$^3$) | 497.00 | Sand(kg/m$^3$) | | 861.90 | | | | | | |
| Super plasticizer (% wt. of binder) | 0.90 | Vol. water/powder | | 1.00 | | | | | | |
| Viscosity modifying agent (% wt. of binder) | 0.20 | Paste composition | | | | | | | | |
| % of air content | 2.00 | kg/m$^3$ | | | | | | | | lit/m$^3$ |
| % of dry material (SP) | 40 | Cement | Fly ash | | Water | | SP | VMA | | Paste |
| % of dry material (VMA) | 40 | 347.90 | 149.10 | | 178.90 | | 4.47 | 0.99 | | 382.60 |

**Table 8.** Mix proportions used and flow-ability performance with variation of HIPS aggregate.

| Cement (kg/m$^3$) | Fly Ash (30% by Wt. of Cement) (kg/m$^3$) | Coarse Aggregate (kg/m$^3$) | | Sand (kg/m$^3$) | HIPS (%) | HIPS (kg/m$^3$) | Flow-Ability | | |
|---|---|---|---|---|---|---|---|---|---|
| | | 12 mm | 10 mm | | | | Slump (mm) | V-Funnel Flow Time (sec) | L-Box (h$_2$/h$_1$) ratio |
| 347.90 | 149.10 | 455.07 | 303.38 | 861.69 | 0 | 0.00 | 598 | 9.80 | 0.83 |
| 347.90 | 149.10 | 455.07 | 303.38 | 776.18 | 10 | 34.48 | 659 | 9.20 | 0.86 |
| 347.90 | 149.10 | 455.07 | 303.38 | 690.13 | 20 | 69.01 | 690 | 8.60 | 0.88 |
| 347.90 | 149.10 | 455.07 | 303.38 | 603.85 | 30 | 103.51 | 723 | 8.00 | 0.89 |
| 347.90 | 149.10 | 455.07 | 303.38 | 517.53 | 40 | 138.00 | 642 | 10.3 | 0.78 |

*2.3. Test Procedure*

2.3.1. Porosity Test on SCC Specimen

All SCC cube specimens of size $100 \times 100 \times 100$ mm were used to determine the porosity. As shown in Figure 5, a porosity test at the curing age of 28 days was conducted according to ASTM C642-13.

Oven-dry mass: The concrete specimen was placed in an oven at 110°C until constant mass was attained. The specimen from the oven was removed and allowed to air dry. The mass of the dried specimen was obtained and designated as *A*.

Saturated specimen mass: The concrete specimen was immersed in water for not less than 48 h till its two successive masses of surface-dried sample values were the same. The determined mass was designated as *B*.

Saturated mass after boiling: The concrete specimen was placed in water for boiling for up to 5 h and allowed to air dry. The mass obtained was designated as *C*.

Immersed apparent mass: The specimen with apparent mass in water was determined using a digital weighing machine and finally this apparent mass of specimen was designated as *D*.

Porosity was calculated as shown in Equations (1)–(3) and tested as given in Figure 5.

$$\text{Dry bulk density} = \rho \left[ \frac{A}{C - D} \right] = g_1 \tag{1}$$

$$\text{Apparent density} = \rho \left[ \frac{A}{A - D} \right] = g_2 \tag{2}$$

$$\text{Volume of permeable pore space voids} = 100 \left[ \frac{g_2 - g_1}{g_2} \right] \tag{3}$$

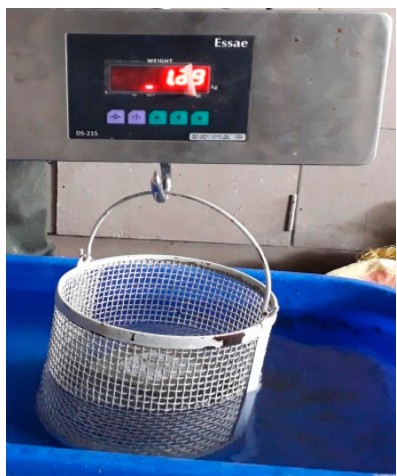

**Figure 5.** Determination of apparent density for porosity test.

### 2.3.2. Water Absorption Test on SCC Specimen

All SCC cube specimens of size 100 mm × 100 mm × 100 mm were tested to determine the water absorption according to ASTM C642-13 [26]. A water absorption test at the curing period of 28 and 90 days was performed as shown in Figure 6. Air dried specimens were placed at 100 °C until a constant weight (x in kg) was obtained. Specimens were immersed in water again and then the saturated surface dried specimens weighted (y in kg). Thus, water absorption (%) was calculated according to Equation (4).

$$\text{Water absorption } (\%) = \frac{100(y-x)}{x} \tag{4}$$

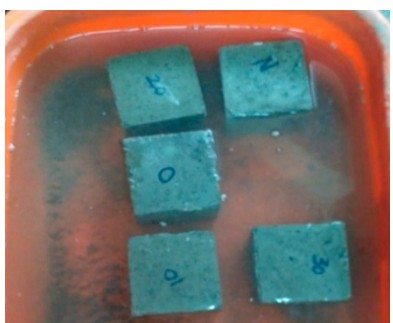

**Figure 6.** SCC specimens with HIPS in water tank.

### 2.3.3. Sorptivity Test on SCC Specimens

A sorptivity test was conducted according to ASTMC1585—13 [27] to determine the rate of water absorption. It measured the change in mass of specimen with respect to time by immersing one surface in a water container to a depth of 3–5 mm. Capillary suction is high during the initial contact of water for unsaturated concrete. Cylindrical specimens of 100 mm diameter and 50 mm depth were sealed with covers except for the bottom surface, and weights were measured. The specimen was immersed in the water container for up to 3–5 mm of the concrete bottom surface. The mass change was recorded from the initial contact of water to different time intervals as given in Table 6. The sorptivity was tested as given in Figure 7a,b and was calculated using Equations (5) and (6). Time intervals with tolerance limits are shown in Table 9.

$$S = \frac{I}{\sqrt{t}} \tag{5}$$

$$I = \frac{m}{\sqrt{t}} \tag{6}$$

where, S = the sorptivity coefficient (mm/$\sqrt{\text{min}}$), $I$= the water absorption per unit concrete surface area (mm), $m_t$ = the change in mass of specimen (grams) w.r.t. time(min), $a$ = the exposed area of the specimen (mm$^2$), $d$ = density of water (g/mm$^3$), and $t$ = time at weight measured (min).

**Table 9.** Sorptivity time intervals according to ASTM C1585—13.

| Time (sec) | Tolerance (+/−) |
|---|---|
| 60 s | 2 s |
| 5 min | 10 s |
| 10 min | 2 min |
| 20 min | 2 min |
| 30 min | 2 min |
| 60 min | 2 min |
| Every hour, up to 6 h measurement on first day | 5 min |
| Once a day, up to 3 days | 2 h |
| Day 4 to 7, three measurements 24 h apart | 2 h |
| Day 7 to 9, one measurement | 2 h |

The initial rate of water absorption (mm/√min) was the slope of the best fit line plotted between *I* vs. (√min), taking values from 1 min to 60 min. The secondary rate of water absorption (mm/√s) was the slope of the best fit line between *I* vs. (√min), taking values from 1 day to 7 days.

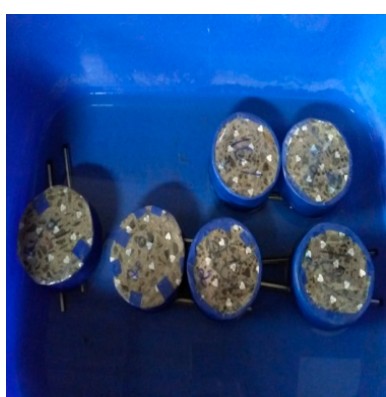

(**a**)

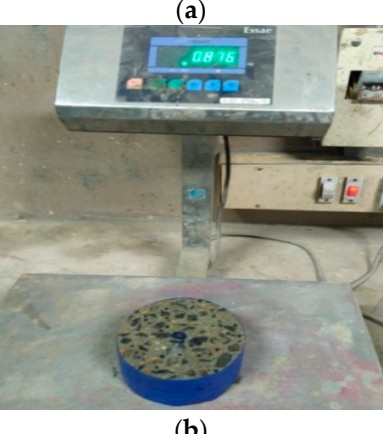

(**b**)

**Figure 7.** (**a**)SCC specimens immersed in water during sorptivity test; (**b**)weighing SCC specimens with exposed bottom surface.

## 3. Result and Discussion

### 3.1. Effect of HIPS Aggregate on Porosity of SCC

Due to rheological enhancement, SCC compacted well with an existing continuous gradation among the matrix and hence porosity reduced up to 30% replacement. Ruiz-Herrero et al. have reported that 200% and 140% higher porosity were identified in concrete at 28 days with 20% of polyethylene (PET) and 20% Poly vinyl chloride (PVC) [28]. Similarly, porosities of conventional

concrete and concrete with coarse electronic waste of 15% PET were 5.35% and 7.65% [11]. In the current investigation, porosity values were obtained of less than 5% for all curing periods. SCC with HIPS exhibited less porosity up to 30% replacement compared to 0% replacement, since HIPS particle size ranging 1 mm–4 mm filled the voids wherein fine aggregates and coarse aggregates couldn't fill voids interlocking among the matrix. Porosity reduced up to 30% HIPS replacement and abruptly changed at 40% in all curing ages, as shown in Figure 8. Hence, a high amount of HIPS replacement from 40% onwards did not help in any property enhancement due to the existence of more plastic per unit volume of concrete, weakening the particle packing density. Water absorption mainly depends on the amount of porosity available in concrete and it was observed that porosity reduced for all curing ages up to 30% of HIPS replacement.

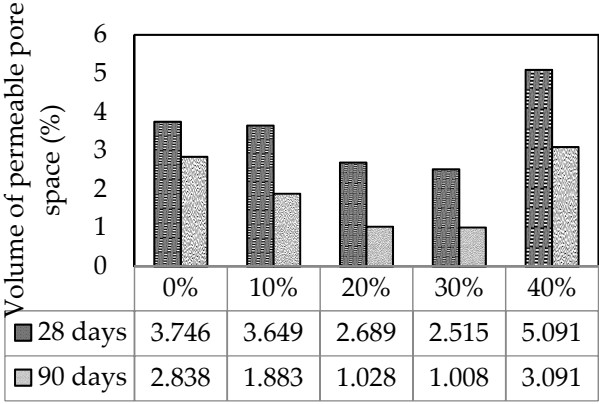

**Figure 8.** Porosity variations with different % of HIPS replacement.

### 3.2. Effect of HIPS Granules in Water Absorption of SCC

Water absorption of conventional concrete and concrete with coarse electronic waste of 15% PET has been found to be 2.12% and 3.56%, respectively [11]. Almost 100% higher water absorption has been noticed in concrete with 15% coarse e-plastic compared to the reference concrete [10]. Coppola et al. have replaced sand with plastic aggregates at 10%, 25%, and 50% in producing lightweight concrete. It has been found that the same amount of absorption as conventional concrete with 10% sand replacement and about 117% higher water absorption was found for 50% sand replacement [29]. Similarly, water absorption of SCC with 15% fine PET aggregate and 30% fly ash has been found to be 7.9% [28]. Water absorption values of SCC specimens after curing of 28 days and 90 days are given in Figure 9. Water absorption reduced with an increase in percentage replacement of HIPS of up to 30%. Water absorption acceptance criteria are shown in Table 10 as per CEB-FIP, 1989 [29].

**Table 10.** Acceptance criteria according to CEB-FIP, 1989.

| Water Absorption (%) | Performance |
| --- | --- |
| >5% | Poor |
| 3%–5% | Average |
| <3% | Good |

Water absorption values of SCC with HIPS after the curing period of 28 days were 3.53%, 3.3%, 3.11%, and 4.96% for 10%, 20%, 30%, and 40%, respectively. The aforesaid values were in between 3%–5% and the performance was found to be average in water absorption at the age of 28 days. Water absorption increased abruptly after 30% replacement of HIPS because minute cracks or voids existed especially at the ITZ, though fly ash acted as filler. However, water absorption with HIPS up to 30% replacement exhibited better performance than the reference concrete absorption value, i.e., 3.89%. The major reason for the reduction in water absorption was the water-repellent nature of HIPS.

Pozzolanic activity resisted the water absorption in longer curing periods, i.e., 90 days. All the test values obtained at 90 days were showing good performance of up to 30% HIPS content, i.e., less than 3%. High volume replacement increased the porosity due to the smooth texture and spherical shape of HIPS compared to natural sand.

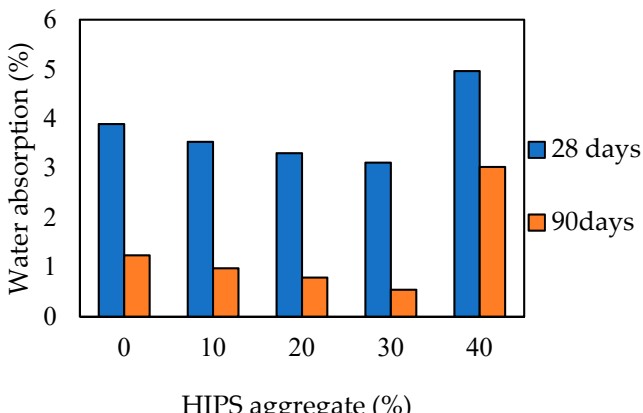

**Figure 9.** Water absorption (%) of SCC mixes with HIPS aggregate (%) at different curing periods.

### 3.3. Effect of HIPS Granuleson Sorptivity of SCC

Sorptivity values of conventional concrete and concrete with coarse electronic waste of 15% PET were obtained as 0.058 and 0.087 mm/$\sqrt{}$min [11]. Sorptivity of concrete with coarse HIPS aggregates (10 to 50%) varied between 0.048 and 0.064 mm/$\sqrt{}$min, 0.0466 and 0.0544 mm/$\sqrt{}$min, and 0.0426 and 0.0515 mm/$\sqrt{}$min for 7, 28, and 90 days, respectively. However, sorptivity values of the control concrete varied from 0.039 mm/$\sqrt{}$min to 0.038 mm/$\sqrt{}$min for 7 to 90 days, respectively [10]. No data exists related to sorptivity values of SCC with fine plastic aggregates and hence it is a valuable attempt to test the sorptivity performance of binary blended SCC with partial fine plastic aggregates. Sorptivity values are shown in Figure 10 at the curing periods of 28 days and 90 days. It was observed that sorptivity values were reduced up to 30% replacement of sand with HIPS aggregate. Reduction in sorptivity was primarily due to the sufficient compaction attained with the enhanced rheology of SCC by pozzolanic fly ash and HIPS. Moreover, the long term pozzolanic reaction of fly ash had a great influence on the reduction of sorptivity at 90 days compared to 28 days. As the curing period increased, the sorptivity values decreased due to reduction of pores at the interfacial zone. From 28 to 90 days, the sorptivity values reduced considerably. Sorptivity values were 0.065, 0.045, 0.034, and 0.028 mm/$\sqrt{}$min for 0%, 10%, 20%, and 30% at 28 days curing and 0.042, 0.036, 0.030, and 0.021 mm/$\sqrt{}$min at 90 days curing from 0%–30% replacement, respectively. For both water absorption and sorptivity testing results showed the same trend line; however, this is not mandatory [10]. The aforesaid results reveal SCC with HIPS is more durable compared to reference SCC concrete.

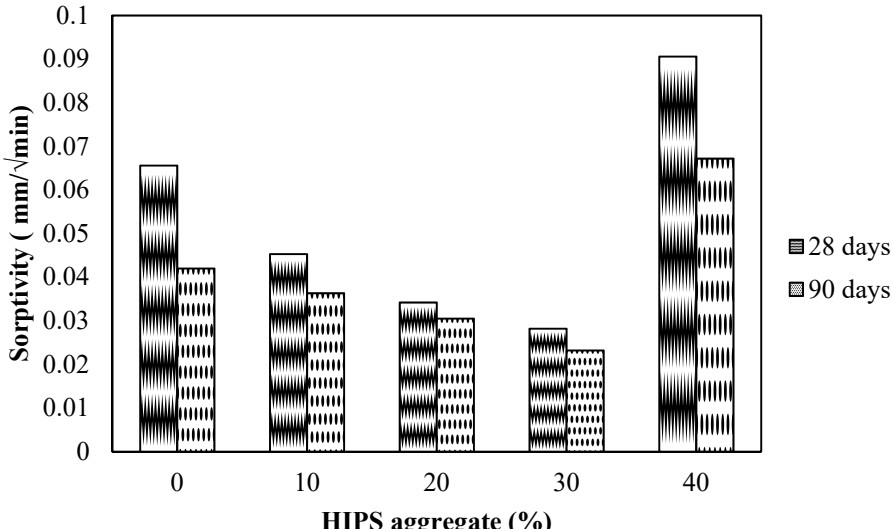

**Figure 10.** Sorptivity of SCC with respect to % variation of HIPS aggregates at 1 h time period.

## 4. Conclusions

The following are the conclusions drawn regarding the combined effects of fly ash and HIPS granules on self-compacting concrete:

- Good compaction was attained with the existence of continuous gradation in the concrete matrix up to 30% HIPS replacement for fine aggregate in SCC. Hence, porosity was reduced about 30% due to enhanced rheology in SCC.
- Performance of HIPS in water absorption was found to be average and satisfactory (<5%). Only 20% of water absorption reduced at 30% of HIPS replacement in place of fine aggregate for SCC compared to reference concrete. Sorptivity values were reduced for concrete with 30% of HIPS replacement of fine aggregate for all curing ages. Reduction of water absorption and sorptivity values was higher for the 90 days compared to the 28 days curing period due to pozzolanic activity.
- The shape, size, and inert nature of HIPS aggregates resemble the properties of sand and it helps for SCC in achieving excellent durable performance. Recycled HIPS can be replaceable up to 30% for natural river sand for producing eco-friendly, durable, and flow-able concrete.

**Author Contributions:** B.R.K.C. and J.P. defined the goals of the study. All authors performed the literature study. All authors have written the manuscript and commented on the final draft.

**Funding:** No specific funding was received for this research.

**Conflicts of Interest:** The authors confirm that there is no conflict of interest.

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
