# Peer review of "Effect of Recycled Plastic Granules as a Partial Substitute for Natural Resource Sand on the Durability of SCC"

_resources, doi:10.3390/resources8030133_

Round 1

Reviewer 1 Report

This paper deals about an interesting investigation and it is worth to be published. However, there are some major issued that should be amended before to be published.

Major issues:

The english is not very bad but a careful reading of the paper is necessary.

In general, the materials must be better described, providing more details, especially in case of aggregates. The information is not very well organized. It is just a mere list of properties.

How the plastic granules are obtained? are they a waste from any industry? Please, be more specific to help the non-familiar readers.

In the paper is mentioned in several times that they are using a self-compacting concrete but there is no any result related with the self-compactability of the concrete. It would be interesting to know how the sand replacement affects the spread flow diameter of SCC, for example.

The analysis of results is very poor. It seems a simple description of results and an analysis with more criticism is necessary, comparing their results with other available in the literature. 

Minor issues:

Line 57: what unit is kg/cum? cum is cubic meter? It is better to express it as kg/m3 as usual.

Line 57: It is hard to understand the sentence: "Fineness modulus of 20mm and 10mm were 6.98 and 5.86, respectively".

Author Response

Thank you very much for your valuable suggestions, guidance and meticulous review of the paper.

Based on reviewer comments updated the entire manuscript. All material properties (Table 1-6 in Section 2.1, pp. 3-5), mix proportions and flow-ability of SCC (Table 4, 5 in section 2.3, pp. 7-8) are tabulated. Especially, HIPS plastic aggregate details are provided with SEM and EDAX analysis (Figure 1,2 in section 2.1, pp.4-5). Result analysis (section 3) is compared with the limited available literature and the research gap (i.e., durability studies on SCC with plastic aggregate as sand is studied up to the possible extent).

FYI, improvements are done in manuscript overall to the best of my knowledge.

Reviewer 2 Report

Title: Effect of recycled plastic granules as a partial substitute for natural resource sand on the durability of SCC

.

This manuscript includes interesting research about reuse of plastic waste. First of all, in my opinion, the topic of the paper is relevant, and the research included in the paper it would have many practical applications and uses.

Comments and Suggestions for Authors

Literature review

The literature review section is mostly acceptable, but it is important to include more recent references (less than 5 years), not only 15 references.

Research Methods

The text needs proofreading as some units must be according to SI  and contractions.

Results and Discussion

Regarding the results, I suggest discussing the results in depth including more references to compare their result with other studies results.

Author Response

Dear Reviewer,

Thank you very much for your valuable suggestions, guidance and meticulous review of the paper.

As per the reviewer suggestions, Introduction section (section 1, pp.1-2) is re-written for the better state-of-the-art. Result section (section 3, pp.10-13) is compared with the available recent literature. Careful proofreading of units is carried out.

FYI, improvements are done in manuscript overall to the best of my knowledge.

Reviewer 3 Report

This paper addressed the fine aggregate of recycled plastic granules with natural resource sand for SCC. Applying some tests to see the properties are considered the major contribution of this paper. This paper in its current state has some question should be addressed:

1. The different of percentages of the HIPS should be addressed in the section 2 for better understanding.

2. As shown in Table 2 and Table 3, the acceptance criteria for curing 28 day or 90 day are not clear.

3. The reference in Table 2, CEB-FIP, 1989, is not shown in the section of reference.

4. The sorptivity values relate to durability should be explained more for better understanding. What is the reference [21] which is not shown?

5. 

Author Response

Dear Reviewer,

Thank you very much for your valuable suggestions, guidance and meticulous review of the paper.

Considering all the valuable suggestions of the reviewer, necessary modifications are done. Details of Mix proportions are clearly mentioned in section 2, pp.8. All the supporting references are properly cited. Result section (section 3, pp.10-13) is compared with the relevant literature available.

FYI, improvements are done in manuscript overall to the best of my knowledge.

Round 2

Reviewer 1 Report

I believe that authors made a good job to reply the reviewer's comments and now the quality of the paper is better. I recommend its publication in present form.

Reviewer 2 Report

I could verify that the suggestions for the paper improvement were included accordingly.

Reviewer 3 Report

This version is improved well. In my opinion, it is qualified for publication.